# Differentially Private Fractional Frequency Moments Estimation with Polylogarithmic Space

**Lun Wang**
UC Berkeley
wanglun@berkeley.edu

**Iosif Pinelis**
Michigan Tech
ipinelis@mtu.edu

**Dawn Song**
UC Berkeley
dawnsong@cs.berkeley.edu

## ABSTRACT

We prove that $\mathbb{F}_p$ sketch, a well-celebrated streaming algorithm for frequency moments estimation, is differentially private as is when $p \in (0, 1]$. $\mathbb{F}_p$ sketch uses only polylogarithmic space, exponentially better than existing DP baselines and only worse than the optimal non-private baseline by a logarithmic factor. The evaluation shows that $\mathbb{F}_p$ sketch can achieve reasonable accuracy with differential privacy guarantee. The evaluation code is included in the supplementary material.

## 1 INTRODUCTION

Counting is one of the most fundamental operations in almost every area of computer science. It typically refers to estimating the cardinality (the $0^{th}$ frequency moment) of a given set. However, counting can actually refer to the process of estimating a broader class of statistics, namely $p^{th}$ frequency moment, denoted $F_p$. Frequency moments estimation is at the core of various important statistical problems. $F_1$ is used for data mining (Cormode et al., 2005) and hypothesis tests (Indyk & McGregor, 2008). $F_2$ has applications in calculating Gini index (Lorenz, 1905; Gini, 1912) and surprise index (Good, 1989), training random forests (Breiman, 2001), numerical linear algebra (Clarkson & Woodruff, 2009; Sarlos, 2006) and network anomaly detection (Krishnamurthy et al., 2003; Thorup & Zhang, 2004). Fractional frequency moments are used in Shannon entropy estimation (Harvey et al., 2008; Zhao et al., 2007) and image decomposition (Geiger et al., 1999).

Non-private frequency moments estimation is systematically studied in the data streaming model (Alon et al., 1999; Charikar et al., 2002; Thorup & Zhang, 2004; Feigenbaum et al., 2002; Indyk, 2006; Li, 2008; Kane et al., 2010; Nelson & Woodruff, 2009; 2010; Kane et al., 2011). This model assumes extremely limited storage such as network routers. The optimal non-private algorithm (Kane et al., 2011) uses only polylogarithmic space to maintain frequency moments. In the present work, we inherit the low space complexity requirement for the versatility of the algorithm.

The data being counted sometimes contains sensitive information. For example, to calculate Gini index, the data should contain pairs of ID and income. Frequency moments of such data, if published, might leak sensitive information. To mitigate, the gold standard of differential privacy (DP) should be applied. Special cases of DP frequency moments estimation such as $p = 0, 1, 2$ are well-studied in a wide spectrum of works (Choi et al., 2020; Smith et al., 2020; Blocki et al., 2012; Sheffet, 2017; Upadhyay, 2014; Choi et al., 2020; Bu et al., 2021; Mir et al., 2011).

In the present work, we make the first customized effort towards DP estimation of fractional frequency moments, *i.e.* $p \in (0, 1]$ with low space complexity. We show that a well-known streaming algorithm, namely $\mathbb{F}_p$ sketch (Indyk, 2006), preserves differential privacy as is. With its small space complexity, $\mathbb{F}_p$ sketch elegantly solves the trilemma between efficiency, accuracy, and privacy.

**Problem Formulation.** We use bold lowercase letters to denote vectors (*e.g.* $\mathbf{a}, \mathbf{b}, \mathbf{c}$) and bold uppercase letters to denote matrices (*e.g.* $\mathbf{A}, \mathbf{B}, \mathbf{C}$). $\{1, \cdots, n\}$ is denoted by $[n]$.

Let $\mathcal{S} = \{(k_1, v_1), \cdots, (k_n, v_n)\}$ $(n \geq 1)$ be a stream of key-value pairs where $k_i \in [m]$ $(m \geq 2), v_i \in [M]$ $(M \geq 1)^1$. We would like to design a randomized mechanism $\mathcal{M}$ that estimates the $p^{th}$

---

[1]In the main text, we consider the cash register model where each update $v_i \geq 0$. The result can be extended to the strict turnstile model as shown in Appendix A.

frequency moment:

$$F_p(\mathcal{S}) = \sum_{k=1}^{m} (\sum_{i=1}^{n} \mathbb{I}(k_i = k)v_i)^p$$

for $p \in (0, 1]$ where $\mathbb{I}$ is an indicator function returning 1 if $k = k_i$ and 0 otherwise.

To provide rigorous privacy guarantee, $\mathcal{M}$ should preserve differential privacy as defined below. In our setting, neighboring data streams differ in one key-value pair.

**Definition 1** (($\epsilon, \delta$)-Differential Privacy)**.** *A randomized algorithm $\mathcal{M}$ is said to preserve ($\epsilon, \delta$)-DP if for two neighboring datasets $\mathcal{S}, \mathcal{S}'$ and any measurable subset of the output space $s$,*

$$\mathbb{P}[\mathcal{M}(\mathcal{S}) \in s] \leq e^\epsilon \mathbb{P}[\mathcal{M}(\mathcal{S}') \in s] + \delta$$

*When $\delta = 0$, we omit it and denote the privacy guarantee as $\epsilon$-DP.*

Oftentimes, $n, m$ is large (*e.g.* IP streams on routers) so $\mathcal{M}$ should take polylogarithmic space in terms of $n, m$.

**Proof Intuition.** We summarize the intuition behind the proof that $\mathbb{F}_p$ sketch is differentially private when $p \in (0, 1]$. Recall that when proving DP for traditional mechanisms such as the Gaussian mechanism, the core is to upper-bound the ratio $\frac{P(x)}{Q(x)}$ where $P(x)$ and $Q(x)$ are the probability density functions of outputs when the inputs are neighboring datasets. In the proof of Gaussian mechanism, $P(x)$ and $Q(x)$ can be viewed as a horizontal translation of each other and the distance between their mean values is the sensitivity of the output.

For $\mathbb{F}_p$, however, neighboring inputs do not translate the output distribution but instead change its scale. For example, when $p = 2$, $P(x)$ and $Q(x)$ are Gaussian distributions with the same mean and different variance. Inspired by the analogy to Gaussian mechanism, we need to address the below two questions to prove differential privacy for $\mathbb{F}_p$ sketches.

> Q1. *How to bound the difference between the scales of $P(x)$ and $Q(x)$?*
>
> Q2. *How to bound the ratio between the density functions of $P(x)$ and $Q(x)$?*

To answer Q1, we propose a new sensitivity definition called *pure multiplicative sensitivity*. Pure multiplicative sensitivity depicts the maximal multiplicative change in the output when the inputs are neighboring datasets. We analyze frequency moments estimation and find that its pure multiplicative sensitivity is approximately $\max\{2^{2p-2}, 2^{2-2p}\}$ when $p \in (0, 1]$ and $n \gg M$.

To answer Q2, we first analyze the special case of $p = 1$. When $p = 1$, $\frac{P(x)}{Q(x)}$ is rigorously upper-bounded and thus $\mathbb{F}_1$ sketch preserves $\epsilon$-DP. By analogy, we conjecture that $\mathbb{F}_p, p \in (0, 1]$ also satisfies similar properties, which is doubly confirmed by the numerically simulated plots in Figure 2. The conjecture is formally proved in Theorem 3.

## 2 RELATED WORK

Frequency moments estimation is thoroughly studied in the data streaming model. Alon et al. (1999) proposed the first space-efficient algorithm for estimating $p^{th}$ frequency moments when $p$ is integer. Indyk (2006) extended the use case from integer moments to fractional moments using stable distributions. A line of following works improve Indyk's algorithm in various aspects such as space complexity (Kane et al., 2010; Nelson & Woodruff, 2009), time complexity (Nelson & Woodruff, 2010; Kane et al., 2011) or accuracy (Li, 2008; 2009).

Several special cases in private frequency moments estimation such as $p = 0, 1, 2$ were also well studied. Choi et al. (2020) and Smith et al. (2020) studied differentially private $F_0$ estimation. They separately proved that the Flajolet-Martin sketch is differentially private as is. Several independent works (Blocki et al., 2012; Sheffet, 2017; Upadhyay, 2014; Choi et al., 2020; Bu et al., 2021) studied the differential privacy guarantee in the special case $p = 2$ under the name of Johnson-Lindenstrauss projection.

On the other hand, there is barely any prior work focusing on differentially private fractional frequency moments estimation. Differentially private distribution estimation algorithms (Acs et al., 2012; Xu

et al., 2013; Bassily & Smith, 2015; Suresh, 2019; Wang et al., 2019) can be used to provide a differentially private estimation of fractional frequency moments. However, they are overkill as their outputs contain much more information than the queried fractional frequency moment. They only provide sub-optimal privacy-utility trade-off and are exponentially worse in terms of space complexity.

Datar et al. (2004) considered a similar (but not the same) mathematical problem to the present work when designing a locality-sensitive hashing scheme. However, their analysis focuses on the simple cases when $p = 1$ and $p = 2$ and totally depends on numerical analysis for $p \in (0, 1)$.

# 3 DIFFERENTIALLY PRIVATE FREQUENCY MOMENTS ESTIMATION

In this section, we first revisit $\mathbb{F}_p$ sketch and then prove the differential privacy guarantee for $\mathbb{F}_p$ sketch step by step. Different from most differential privacy analyses based on additive sensitivity, our proof depends on a variant of the multiplicative sensitivity (Dwork et al., 2015) called *pure multiplicative sensitivity*. We give the first analysis of pure multiplicative sensitivity for $p$-th frequency moments. Then we motivate the differential privacy proof using a special case when $p = 1$. Finally we proceed to the general proof that $\mathbb{F}_p$ sketch preserves differential privacy. The main challenge stems from the fact that the density functions of $p$-stable distributions have no close-form expressions when $p \in (0, 1)$.

## 3.1 REVISITING $\mathbb{F}_p$ SKETCH

For completeness, we revisit the well-celebrated $\mathbb{F}_p$ sketch by Indyk (2006) (also known as stable projection or compressed counting). We first introduce $p$-stable distribution, the basic building block in $\mathbb{F}_p$ sketch. Then we review how to construct and query $\mathbb{F}_p$ sketch using stable distributions.

**Definition 2** ($p$-stable distribution). *A random variable $X$ follows a $\beta$-skewed $p$-stable distribution if its characteristic function is*

$$\phi_X(t) = \exp(-\zeta|t|^p(1 - \sqrt{-1}\beta \operatorname{sgn}(t)\tan(\frac{\pi p}{2})))$$

*where $-1 \leq \beta \leq 1$ is the skewness parameter, $\zeta > 0$ is the scale parameter to the $p^{th}$ power.*

In this paper, we focus on stable distributions with $\beta = 0$, namely symmetric stable distributions. We denote a symmetric $p$-stable distribution by $\mathcal{D}_{p,\zeta}$, and slightly abuse the notation to denote the density function as $\mathcal{D}_{p,\zeta}(x)$. Note that the density function is the inverse Fourier transform of the characteristic function.

$$\mathcal{D}_{p,\zeta}(x) = \frac{1}{2\pi}\int_{\mathbb{R}}\exp(-\sqrt{-1}tx)\phi(t)dt = \frac{1}{2\pi}\int_{\mathbb{R}}\cos(xt)\exp(-\zeta|t|^p)dt$$

If two independent random variables $X_1, X_2 \sim \mathcal{D}_{p,1}$, then $C_1X_1 + C_2X_2 \sim \mathcal{D}_{p,C_1^p+C_2^p}$. We refer to this property as $p$-*stability*. $\mathbb{F}_p$ sketch leverages the $p$-stability of these distributions to keep track of the frequency moments.

The pseudo-code for vanilla $\mathbb{F}_p$ sketch is presented in Algorithm 1. To construct, a sketch of size $r$ is initialized to all zeros and a projection matrix $\mathbf{P}$ is sampled from $\mathcal{D}_{p,1}^{r \times m}$ (line 2). For each incoming key-value pair $(k_i, v_i)$, we multiply the one-hot encoding of $k_i$ scaled by $v_i$ with the projection matrix $\mathbf{P}$ and add it to the sketch (line 4).

$$\mathbf{a} = \sum_{i=1}^{n}\mathbf{P} \times v_i\mathbf{e}_{k_i} = \sum_{k=1}^{m}\mathbf{P} \times (\sum_{k_i=k}v_i)\mathbf{e}_{k_i} \sim \mathcal{D}_{p,F_p(\mathcal{S})}^{r}$$

To query the sketch, we estimate $\zeta$ from $\mathbf{a}$ using various estimators such as median, inter-quantile, geometric mean or harmonic mean as suggested by Indyk (2006), Li (2008) and Li (2009).

---

**Algorithm 1:** $\mathbb{F}_p$ sketch.

---

**Input** : Data stream: $\mathcal{S} = \{(k_1, v_1), \cdots, (k_n, v_n)\}$

1  **Construct:**
2    |   Initialize $\mathbf{a} = \{0\}^r$, $\mathbf{P} \sim \mathcal{D}_{p,1}^{r \times m}$;
3  **Update:**
4    |   **for** $i \in [n]$ **do** Let $e_{k_i}$ be the one-hot encoder of $k_i$, $\mathbf{a} = \mathbf{a} + \mathbf{P} \times v_i \mathbf{e}_{k_i}$ ;
5  **Query:**
6    |   return scale_estimator($\mathbf{a}$);

---

### 3.2   PURE MULTIPLICATIVE SENSITIVITY OF FREQUENCY MOMENTS ESTIMATION

As we will see in the following two subsections, the differential privacy proof for $\mathbb{F}_p$ sketch depends on the pure multiplicative sensitivity of $p$-th frequency moments. As the first step, we give the definition of pure multiplicative differential privacy. "Pure" is to distinguish from multiplicative sensitivity as defined in Dwork et al. (2015).

**Definition 3** (Pure multiplicative sensitivity). *The multiplicative sensitivity of a function $\mathcal{M}$ is defined as the maximum ratio between outputs on neighboring inputs $\mathcal{S}$ and $\mathcal{S}'$.*

$$\rho_p(n) = \sup_{|\mathcal{S}|=n, |\mathcal{S}'|=n, d(\mathcal{S},\mathcal{S}')=1} \big| \frac{\mathcal{M}(\mathcal{S})}{\mathcal{M}(\mathcal{S}')} \big|$$

*We might omit the subscript and argument when they are clear from the context.*

The pure multiplicative sensitivity of $F_p$ is as below.

**Theorem 1** (Multiplicative sensitivity of $F_p$). *A mechanism $\mathcal{M}$ which calculates $F_p, p \in (0, 1]$ has pure multiplicative sensitivity upper bounded by*

$$\rho_p \leq 2^{2-2p} \big( \frac{n - 1 + M}{n - 1 + (m-1)^{\frac{p-1}{p}}} \big)^p$$

*Proof for Theorem 1.* Theorem 1 gives an upper bound on the multiplicative change when two input datasets with the same size $m$ differ in one entry. To prove, we first consider a slightly different setting when the second dataset is generated by adding an entry to the first dataset. Then the neighboring datasets in the original setting can be generated by adding different entries to the same dataset. Thus, taking the division of the upper and lower bound of the sensitivity in the incremental setting will give an upper bound for sensitivity in the original setting.

Concretely, let $\mathbf{u} = \{u_1, \cdots, u_m\}$ where $u_i > 0$, $\sum_{i=1}^m u_i = s$, $\Delta \geq 0$. We would like to find both upper and lower bounds for the below expression.

$$\frac{\sum_{i=2}^m u_i^p + (u_1 + \Delta)^p}{\sum_{i=2}^m u_i^p + u_1^p}, \forall p \in (0, 1] \tag{1}$$

To bound expression (1), we first observe the following two inequalities (2) and (3).

$$\forall a, b, c, d > 0, a \geq b, c \geq d, \frac{a + c}{a + d} \leq \frac{b + c}{b + d}. \tag{2}$$

$$\forall p \in (0, 1], (\sum_{i=1}^m u_i)^p \leq \sum_{i=1}^m u_i^p \leq m^{1-p} (\sum_{i=1}^m u_i)^p \tag{3}$$

Inequality (2) can be proved with simple algebra. The left-hand-side of inequality (3) follows because $\sum_1^m u_i^p$ is concave in $(u_1, \ldots, u_n)$ in the simplex defined by the conditions $u_i \geq 0$ for all

$i$, and $\sum_1^m u_i = s$ and hence the minimum of $\sum_1^m u_i^p$ on the simplex is attained at a vertex of the simplex. The right-hand-side of inequality (3) is an instance of the well-known generalized mean inequality (Sỳkora, 2009) or Hölder inequality (Hölder, 1889).

First, let's upper bound expression (1). According to inequality (2) and (3),

$$\frac{\sum_{i=2}^m u_i^p + (u_1 + \Delta)^p}{\sum_{i=2}^m u_i^p + u_1^p} \overset{(2)+(3)}{\leq} \frac{(\sum_{i=2}^m u_i)^p + (u_1 + \Delta)^p}{(\sum_{i=2}^m u_i)^p + u_1^p} = \frac{(s - u_1)^p + (u_1 + \Delta)^p}{(s - u_1)^p + u_1^p} \overset{(3)}{\leq} 2^{1-p}(1 + \frac{\Delta}{s})^p$$

Similarly, to lower bound expression (1),

$$\frac{\sum_{i=2}^m u_i^p + (u_1 + \Delta)^p}{\sum_{i=2}^m u_i^p + u_1^p} \overset{(2)+(3)}{\geq} \frac{(m-1)^{1-p}(\sum_{i=2}^m u_i)^p + (u_1 + \Delta)^p}{(m-1)^{1-p}(\sum_{i=2}^m u_i)^p + u_1^p}$$

$$= \frac{(s - u_1)^p + ((m-1)^{\frac{p-1}{p}}(u_1 + \Delta))^p}{(s - u_1)^p + ((m-1)^{\frac{p-1}{p}} u_1)^p}$$

$$\overset{(3)}{\geq} 2^{p-1}\left(\frac{((m-1)^{\frac{p-1}{p}} - 1)u_1 + s + (m-1)^{\frac{p-1}{p}}\Delta}{((m-1)^{\frac{p-1}{p}} - 1)u_1 + s}\right)^p$$

$$\geq 2^{p-1}(1 + \frac{(m-1)^{\frac{p-1}{p}}\Delta}{s})^p$$

Taking the division between the supremum and the infimum, we get

$$\rho_p \leq 2^{2-2p}\left(\frac{s + M}{s + (m-1)^{\frac{p-1}{p}}}\right)^p \leq 2^{2-2p}\left(\frac{n - 1 + M}{n - 1 + (m-1)^{\frac{p-1}{p}}}\right)^p \qquad \square$$

In a typical streaming model where $m$ is large and $n \gg M$, $\rho_p \lessapprox 2^{2-2p} \leq 4$. To get a better sense of how $\rho$ changes with $p$, we plot several curves with different hyper-parameters in Figure 1. Note that the pure multiplicative sensitivity only depends on $n, m, M$ and $p$ which are public information.

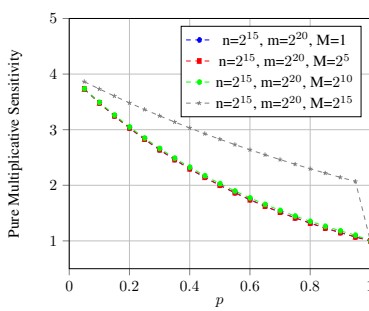

Figure 1: Pure multiplicative sensitivity.

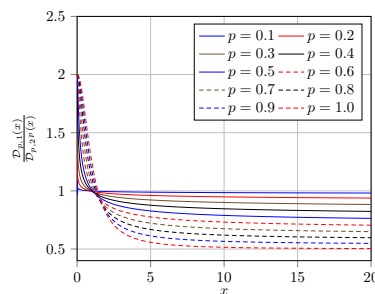

Figure 2: The curves of $\frac{\mathcal{D}_{p,1}(x)}{\mathcal{D}_{p,2^p}(x)}$ with different values of $p \in (0, 1]$ on $\mathbb{R}^+$. The negative half is symmetric.

### 3.3 Differentially Private $\mathbb{F}_1$ Sketch

Instead of directly diving into the complete analysis, we first motivate the analysis with the special case of $p = 1$. In this case, the symmetric $1^{st}$-stable distribution is the well-known Cauchy distribution: $\mathcal{D}_{1,\zeta}(x) = \frac{1}{\pi} \cdot \frac{\zeta}{\zeta^2 + x^2}$, and thus the analyses are significantly simplified. The main purpose of this section is to pave the way for the proof of general $\mathbb{F}_p$ sketch.

**Theorem 2** ($\epsilon$-DP for $\mathbb{F}_1$ sketch). *Let $\rho_1$ represent the multiplicative sensitivity of the first frequency moments. When the size of the sketch $r = 1$, $\mathbb{F}_1$ is $\ln \rho_1$-differentially private.*

*Proof for Theorem 2.* $\frac{\mathcal{D}_{1,F_1}(x)}{\mathcal{D}_{1,\rho_1 F_1}(x)} = \frac{\rho_1^2 F_1^2 + x^2}{\rho_1(F_1^2 + x^2)}$ is a decreasing function of $x$ when $x \in (0, \infty)$ because its derivative $\frac{2(\rho_1 - \rho_1^3)F_1^2 x}{\rho_1^2(F_1^2 + x^2)^2}$ is non-positive. Thus,

$$\frac{1}{\rho_1} = \frac{\mathcal{D}_{1,F_1}(\infty)}{\mathcal{D}_{1,\rho_1 F_1}(\infty)} \leq \frac{\mathcal{D}_{1,F_1}(x)}{\mathcal{D}_{1,\rho_1 F_1}(x)} \leq \frac{\mathcal{D}_{1,F_1}(0)}{\mathcal{D}_{1,\rho_1 F_1}(0)} = \rho_1$$

Then, for any data stream $\mathcal{S}$ and arbitrary measurable subset $s$,

$$\mathbb{P}[\mathbb{F}_1(\mathcal{S}) \in s] = \int_{x \in s} \mathcal{D}_{1,F_1(\mathcal{S})}(x)dx = \int_{x \in s} \frac{\mathcal{D}_{1,F_1(\mathcal{S})}(x)}{\mathcal{D}_{1,F_1(\mathcal{S}')}(x)} \mathcal{D}_{1,F_1(\mathcal{S}')}(x)dx$$

$$\leq \int_{x \in s} \rho_1 \mathcal{D}_{1,F_1(\mathcal{S}')}(x)dx = e^{\ln \rho_1} \mathbb{P}[\mathbb{F}_1(\mathcal{S}') \in s] \qquad \square$$

### 3.4 DIFFERENTIALLY PRIVATE $\mathbb{F}_p$ SKETCH, $p \in (0,1]$

The example of $\mathbb{F}_1$ being $\epsilon$-DP indicates the possibility that $\mathbb{F}_p$ might have similar property when $p \in (0,1]$. To validate, we plot the curves for different values of $p$s as shown in Figure 2. From the figure we can tell that when $p \in (0,1]$, the ratio $\frac{\mathcal{D}_{p,1}(x)}{\mathcal{D}_{p,2}(x)}$ seems to be well-bounded and preserve $\epsilon$-DP.

We now prove the conjecture as formalized in Theorem 3.

**Theorem 3** ($\epsilon$-DP for Algorithm 1). *Let $\rho_p$ represent the multiplicative sensitivity of the p-th frequency moments. When $r = 1$ and $p \in (0,1]$, $\mathbb{F}_p$ sketch (Algorithm 1) is $\frac{1}{p} \ln \rho_p$-differentially private.*

*Proof for Theorem 3.* To prove Theorem 3, we prove the following inequality.

$$\rho_p^{-\frac{1}{p}} < \rho_p^{-1} \leq \frac{\mathcal{D}_{p,F_p}(x)}{\mathcal{D}_{p,\rho_p F_p}(x)} \leq \rho_p^{\frac{1}{p}}$$

We first prove the right-hand-side of the inequality. Observe that $\zeta^{-\frac{1}{p}} \mathcal{D}_{p,1}(\zeta^{-\frac{1}{p}} x) = \frac{\zeta^{-\frac{1}{p}}}{2\pi} \int_{\mathbb{R}} \cos(\zeta^{-\frac{1}{p}} xt) \exp(-|t|^p)dt \overset{\star}{=} \frac{1}{2\pi} \int_{\mathbb{R}} \cos(xt) \exp(-\zeta|t|^p)dt = \mathcal{D}_{p,\zeta}(x)$ where $\star$ substitutes $t$ with $\zeta^{\frac{1}{p}} t$ using integration by substitution. Thus,

$$\frac{\mathcal{D}_{p,F_p}(x)}{\mathcal{D}_{p,\rho_p F_p}(x)} = \rho_p^{\frac{1}{p}} \frac{\mathcal{D}_{p,1}(F_p^{-\frac{1}{p}} x)}{\mathcal{D}_{p,1}((\rho_p F_p)^{-\frac{1}{p}} x)} \leq \rho_p^{\frac{1}{p}} \frac{\mathcal{D}_{p,F_p}(0)}{\mathcal{D}_{p,\rho_p F_p}(0)} = \rho_p^{\frac{1}{p}}$$

as $\mathcal{D}_{p,1}$ is increasing on $(-\infty, 0]$ and decreasing on $[0, \infty)$, and $\rho_p \geq 1$.

To prove the left-hand-side of the inequality, we reorganize it into the format of a Fourier transform.

$$\int_0^\infty (\rho_p \exp(-F_p t^p) - \exp(-\rho_p F_p t^p)) \cos(tx)dt \geq 0$$

It suffices to show that

$$h(\rho) = \int_0^\infty \frac{\exp(-\rho F_p t^p)}{\rho} \cos(tx)dt$$

is decreasing. Taking the first derivative of $h$, we have

$$\frac{\partial h}{\partial \rho} = -\frac{1}{\rho^2} \int_0^\infty g(t) \cos(tx)dt, \text{ where } g(t) = \exp(-\rho F_p t^p)(\rho F_p t^p + 1)$$

According to Pólya criterion (Gneiting, 2001), it suffices to show that $g$ is positive definite. We first observe that the function $0 \leq u \mapsto (1 + u^{1/2})e^{-u^{1/2}}$ is the Laplace transform (Schiff, 1999) of the positive function $0 < t \mapsto \dfrac{e^{-1/(4t)}}{4\sqrt{\pi}\, t^{5/2}}$ (the proof is deferred to the end) and hence a mixture of exponential functions $0 \leq u \mapsto e^{-cu}$ with $c > 0$. Thus with variable substitution, the function $s \mapsto (1 + |s|^p)e^{-|s|^p}$ is a mixture of functions $s \mapsto e^{-c|s|^{2p}}$ with $c > 0$, which are positive definite for any $p \in (0,1]$ as they are characteristic functions of stable distributions.

The last step is to prove the function $F(u) = (1 + u^{1/2})e^{-u^{1/2}}, u \geq 0$ is the Laplace transform of $f(t) = \dfrac{e^{-1/(4t)}}{4\sqrt{\pi}\, t^{5/2}}, t > 0$: $F(u) = \int_0^\infty f(t)e^{-ut}dt, u \geq 0$. Let $R(u) = F(u)$ and $L(u) = \int_0^\infty f(t)e^{-ut}dt$. We observe that $\lim_{x \to \infty} L(x) = \lim_{x \to \infty} R(x) = \lim_{x \to \infty} L'(x) =$

$\lim_{x \to \infty} R'(x) = 0$ so it is enough to show that $L''(u) = R''(u)$. After a simple rescaling, it is enough to show

$$J(a) := \int_0^\infty \exp\left\{ -\frac{1}{t} - at \right\} \frac{dt}{2\sqrt{t}} = \frac{\sqrt{\pi}}{2} \frac{e^{-2\sqrt{a}}}{\sqrt{a}}$$

where $a > 0$ as both sides do not contain linear terms. Using substitutions $t = u^2$ and then $u = 1/(x\sqrt{a})$, we get

$$J(a) = \int_0^\infty \exp\left\{ -\frac{1}{u^2} - au^2 \right\} du = K(a)/\sqrt{a},$$

where $K(a) := \int_0^\infty \exp\left\{ -ax^2 - \frac{1}{x^2} \right\} \frac{dx}{x^2}$. Note that $K'(a) = -J(a)$ and $K(a) = J(a)\sqrt{a}$. So, we get the differential equation $J'(a) = -\left( \frac{1}{\sqrt{a}} + \frac{1}{2a} \right) J(a)$, whose general solution is given by $J(a) = \frac{c}{\sqrt{a}} e^{-2\sqrt{a}}$ for a constant $c$. To determine $c$, note that $K(a) = J(a)\sqrt{a} = \int_0^\infty \exp\left\{ -\frac{1}{u^2} - au^2 \right\} du \sqrt{a} = \int_0^\infty \exp\left\{ -\frac{a}{y^2} - y^2 \right\} dy$ and $c = K(0+) = \int_0^\infty \exp\{-y^2\} dy = \frac{\sqrt{\pi}}{2}$. $\qquad \square$

### 3.5 PRIVACY AMPLIFICATION BY SUB-SAMPLING

The last step of Algorithm 1 estimates $\zeta$ given samples from the stable distributions. There are many candidate estimators such as the geometric estimator and the harmonic estimator (Li, 2008; 2009). These estimators typically, as suggested in Li (2008), require at least $r \geq 50$ samples to give an accurate estimation of $\zeta$. However, the privacy parameter $\epsilon$ grows linearly with $r$ with trivial composition (Dwork et al., 2006), which might result in too weak privacy protection.

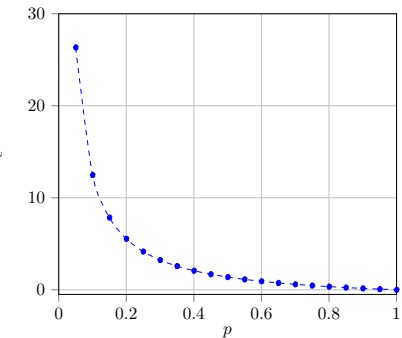

Figure 3: Privacy budget $\epsilon$ vs. $p$. $n = 2^{15}, m = 2^{20}, M = 2^4$.

To address, we follow the standard approach, amplifying privacy using sub-sampling. Different from Algorithm 1, each input has probability $q$ to be inserted into each dimension of $\mathbf{a}$, as presented in Algorithm 2. If we take $q = \frac{1}{r}$, then the privacy parameters in Theorem 3 hold as is. The proof is a simple application of the composition theorems (Dwork et al., 2006) and privacy amplification (Balle et al., 2018).

**Theorem 4** ($\epsilon$-DP for Algorithm 2). *Let $\rho_p$ represent the multiplicative sensitivity of the p-th frequency moments. When $p \in (0, 1]$, $\mathbb{F}_p$ sketch with sub-sampling rate $q$ is $\frac{qr}{p} \ln \rho_p$-differentially private.*

---

**Algorithm 2:** $\mathbb{F}_p$ sketch with sub-sampling. The only change appears in line 3-4 and 7, corresponding to line 3 and 5 in Algorithm 1. Bernoulli($q$) refers to Bernoulli distribution with success probability $q$.

**Input** : Data stream: $\mathcal{S} = \{(k_1, v_1), \cdots, (k_n, v_n)\}$
1 **Construction**
2     Initialize $\mathbf{a} = \{0\}^r$, $P \sim \mathcal{D}_{p,1}^{r \times m}$;
3     **for** $i \in [n]$ **do**
4        $b \sim$ Bernoulli($q$);
5        Let $e_{k_i}$ be the one-hot encoder of $k_i$, $\mathbf{a} = \mathbf{a} + \mathbf{P} \times bv_i\mathbf{e}_{k_i}$
6 **Query**
7     return scale_estimator($\mathbf{a}$)$/q^p$;

---

### 3.6 UTILITY OF ALGORITHM 2

We depict the accuracy of a $F_p$ estimator with a pair of parameters $(\gamma, \eta)$.

**Definition 4** ($(\gamma, \eta)$-Accuracy). *A randomized algorithm $\mathcal{M}$ is said to be $(\gamma, \eta)$-accurate if*

$$(1 - \gamma)F_p(\mathcal{S}) \leq \mathcal{M}(\mathcal{S}) \leq (1 + \gamma)F_p(\mathcal{S}) \quad w.p. \quad 1 - \eta$$

Algorithm 2 satisfies the following utility guarantee. The space complexity is only worse than the optimal non-private algorithm (Kane et al., 2011) by a logarithmic factor. The accuracy bound is also a worst-case bound and the performance in practice is typically much better (Section 4).

**Theorem 5** (Utility of Algorithm 2). $\forall p \in (0, 1]$ and $\forall \gamma, \eta \in (0, 1)$, Algorithm 2 is $(\gamma + \sqrt{\frac{q^p - q^{2p}}{\lambda}}, \eta + \lambda)$-accurate if $r = \mathcal{O}\big(\gamma^{-2} \log(\frac{1}{\eta})\big)$. In this case, Algorithm 2 uses $\mathcal{O}\big(\gamma^{-2} \log(mM/(\gamma\eta)) \log(\frac{1}{\eta})\big)$ bits.

*Proof.* Let $\mathcal{SA}_q(\cdot)$ represent the sub-sampling process and $\mathbb{F}_p^r$ represent a $\mathbb{F}_p$ sketch with length $r$. Then Algorithm 2 can be represented as $\mathbb{F}_p^r \circ \mathcal{SA}_q$ where $\circ$ represents composition of mechanisms.

First, we need the accuracy of $\mathbb{F}_p$ sketch. According to Theorem 4 of Indyk (2006), if we fix the sub-sampled items,

$$\mathbb{P}[|\mathbb{F}_p^{\mathcal{O}\big(\gamma^{-2} \log(\frac{1}{\eta})\big)} \circ \mathcal{SA}_q(\mathcal{S}) - F_p \circ \mathcal{SA}_q(\mathcal{S})| \leq \gamma F_p \circ \mathcal{SA}_q(\mathcal{S})] \geq 1 - \eta$$

Second, we need the accuracy of the sub-sampling process. The expectation and variance of the sub-sampling process is as follow.

$$\mathbb{E}[F_p \circ \mathcal{SA}(\mathcal{S})] = q^p F_p(\mathcal{S}) \tag{4}$$

$$\begin{aligned}
\mathbb{V}[F_p \circ \mathcal{SA}(\mathcal{S})] &= \mathbb{E}[(F_p \circ \mathcal{SA}(\mathcal{S}))^2] - \mathbb{E}^2[F_p \circ \mathcal{SA}(\mathcal{S})] \\
&\leq F_p(\mathcal{S}) \times \mathbb{E}[F_p \circ \mathcal{SA}(\mathcal{S})] - q^{2p} F_p^2(\mathcal{S}) = (q^p - q^2 p) F_p(\mathcal{S})
\end{aligned} \tag{5}$$

According to Chebyshev's inequality,

$$\mathbb{P}[|F_p \circ \mathcal{SA}(\mathcal{S}) - q^p F_p(\mathcal{S})| \leq \sqrt{\frac{q^p - q^{2p}}{\lambda}} F_p(\mathcal{S})] \geq 1 - \lambda \tag{6}$$

Combining (4), (6) and (6) we get Theorem 5. $\qquad\square$

# 4 EVALUATION

## 4.1 EVALUATION SETUP

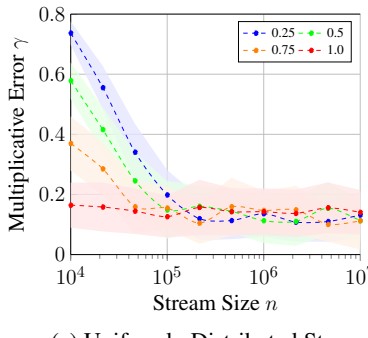
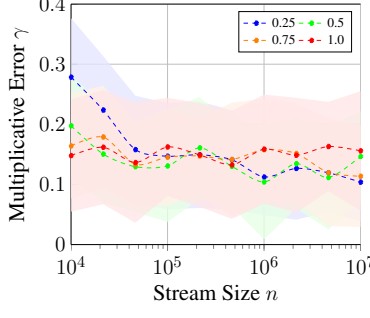

(a) Uniformly Distributed Stream.  (b) Binomially Distributed Stream.

Figure 4: Results on Synthetic Data.

As we would like to empirically understand $\mathbb{F}_p$ sketch's trade-off between space, error and privacy, we evaluate $\mathbb{F}_p$ with $p \in \{0.25, 0.5, 0.75, 1\}$ using synthetic streams of different sizes and distributions. We also evaluate $\mathbb{F}_p$ with $p \in \{0.05, 0.1, \cdots, 0.95, 1\}$ on real-world data. All the experiments were run on a Ubuntu18.04 LTS server with 32 AMD Opteron(TM) Processor 6212 with 512GB RAM.

**Synthetic Data.** We first evaluate $\mathbb{F}_p$ sketches using synthetic data. We synthesize two kinds of data: the key domain is either uniformly or binomially distributed. The value domain is $\{1\}$ by default. The size of the key domain is 1000.

**Real-world Data.** We also evaluate $\mathbb{F}_p$ sketches using real-world application usage data (Ye et al., 2019) collected by TalkingData SDK. There are more than 30 million events in this dataset, each representing one access to the TalkingData SDK. We view the event type as the key and the value is set to 1 by default.

## 4.2 EVALUATION RESULTS

In this section, we present the evaluation results. To avoid the influence of outliers, we report the median and interquartile of 100 runs for each data point except for the real-data evaluation. For all the evaluation, the sketch size $r$ is 50 as suggested in Li (2008). The sub-sampling rate in all the experiments is 0.02.

**Synthetic Data.** The evaluation results on synthetic data are presented in Figure 4. For uniformly distributed data, we observe that as the stream size increases, the multiplicative error decreases. We conjecture the reason to be the effect of sub-sampling. Concretely, each bin in the value domain has to get enough samples to approximate the behavior of the true distribution. On the other hand, when the data is binomially distributed, the multiplicative error is relatively stable with small fluctuation. We conjecture the reason is that as binomial distribution is more concentrated, the sample complexity is smaller than uniform distribution. Besides, for uniformly distributed data, $p$s close to 0 have relatively large errors while the errors when $p$ is close to 1 are small. The reason is that the further $p$ is from 1, the larger the influence of sub-sampling.

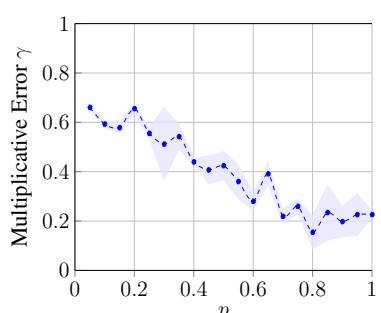

Figure 5: Results on Real-world Data.

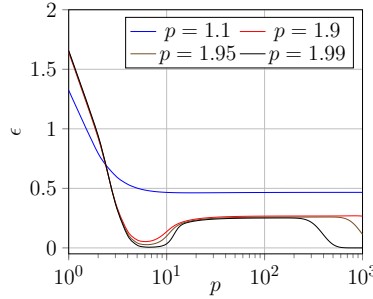

Figure 6: The curves of $\frac{\mathcal{D}_{p,1}(x)}{\mathcal{D}_{p,2^p}(x)}$ with different values of $p \in (1,2)$ on $\mathbb{R}^+$. The negative half is symmetric. The x-axis is log-scale to highlight the complex monotone trends.

**Real-world Data.** The evaluation results for real-world data are presented in Figure 5. We sampled 100,0000 data points from the dataset and the key has a domain of size 1488095. Each data point is the median of 5 runs. We observe that the further $p$ is from 1, the higher the multiplicative error. This conforms with our observation in the evaluation on synthetic data.

## 5 CONCLUSION & FUTURE WORK

This paper takes an important step towards narrowing the gap of space complexity between private and non-private frequency moments estimation algorithms. We prove that $\mathbb{F}_p$ is differentially private as is when $p \in (0,1]$ and thus give the first differentially private frequency estimation protocol with polylogarithmic space complexity.

At the same time, we observe several open challenges. First, the proof does not easily extend to $p \in (1,2)$. Figure 6 exhibits the complexity of monocity of $\frac{\mathcal{D}_{p,1}(x)}{\mathcal{D}_{p,2^p}(x)}$ when $p \in (1,2)$. The most complex curve when $p = 1.99$ is composed of three monotonic parts in the figure. Hence, an interesting next step is to fully understand the monotonicity pattern of the ratio curve when $p \in (1,2)$ and get corresponding privacy parameters. Second, the space complexity of Algorithm 2 is still worse than the optimal non-private algorithm by a factor of $\log(m)$. It is interesting to check whether the optimal algorithm (Kane et al., 2011) also preserves differential privacy.

**Ethics Statement.** Our work study the intrinsic differential privacy of $\mathbb{F}_p$ sketches. $\mathbb{F}_p$ sketches should be used with careful calibration to make sure the output is accurate and provides reasonable privacy guarantee. All the datasets and packages used are open-sourced.

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

# A   PURE MULTIPLICATIVE SENSITIVITY IN STRICT TURNSTILE MODEL

In this section, we derive the pure multiplicative sensitivity of $F_p$ in the strict turnstile model. In the strict turnstile model, for a key-value stream $\mathcal{S} = \{(k_1, v_1), \cdots, (k_n, v_n)\}$ $(n \geq 1)$ where $k_i \in [m]$ $(m \geq 2), v_i \in \{-M, \cdots, M\}$ $(M \geq 1)$, the sum of $v$s of the same key should always be non-negative:

$$\sum_{i=1}^{n} \mathbb{I}(k_i = k) v_i \geq 0$$

Besides, for the utility of the result, we need to assume that $M < n - 1$.

**Theorem 6** (Multiplicative sensitivity of $F_p$ in strict turnstile model). *A mechanism $\mathcal{M} < n - 1$ which calculates $F_p, p \in (0, 1]$ in the strict turnstile model when has pure multiplicative sensitivity upper bounded by*

$$\rho_p^{st} \leq 2^{2-2p}(1 + \frac{2M}{n - 1 - M})^p$$

*Proof for Theorem 6.* An upper bound for the sensitivity of $F_p$ in the strict turnstile model can be derived by taking the division of the upper and lower bound in the incremental setting following the same logic as the proof for Theorem 1. The upper bound is the same as in the proof of Theorem 6 so we only need to calculate the lower bound of the following expression.

First, we observe the following two inequalities.

$$\forall a, b, d > 0, c \geq 0, a \leq b, c \leq d, \frac{a + c}{a + d} \leq \frac{b + c}{b + d}. \tag{7}$$

$$\frac{\sum_{i=2}^{m} u_i^p + (u_1 - \Delta)^p}{\sum_{i=2}^{m} u_i^p + u_1^p} \overset{(3)+(7)}{\geq} \frac{(\sum_{i=2}^{m} u_i)^p + (u_1 - \Delta)^p}{(\sum_{i=2}^{m} u_i)^p + u_1^p}$$
$$= \frac{(s - u_1)^p + (u_1 - \Delta)^p}{(s - u_1)^p + u_1^p}$$
$$\overset{(3)}{\geq} 2^{p-1}(1 - \frac{\Delta}{s})^p$$

Taking the division between the supremum and the infimum, we get

$$\rho_p^{st} \leq 2^{2-2p}(1 + \frac{2\Delta}{s - \Delta})^p \leq 2^{2-2p}(1 + \frac{2M}{n - 1 - M})^p \qquad \square$$

As shown in Figure 7, when $m$ is the same, the sensitivity is very close to the sensitivity in the cash register model if $M \ll n$.

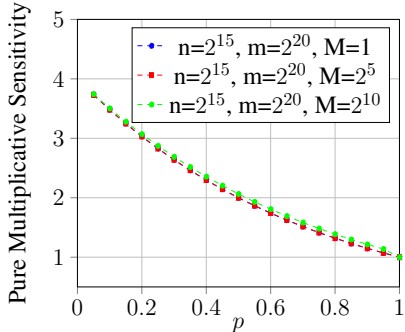

Figure 7: Pure multiplicative sensitivity in the Strict Turnstile Model.

