# OpenReview forum: "Differentially Private Fractional Frequency Moments Estimation with Polylogarithmic Space"
_ICLR.cc/2022/Conference — ICLR 2022 Poster_

### Official Review · Reviewer_Pbv7 · 2021-10-26

**Correctness:** 3
**Technical Novelty And Significance:** 2
**Empirical Novelty And Significance:** 3
**Recommendation:** 6
**Confidence:** 4

**Main Review:**

**Updated my review to be more positive**

Strengths: They provide reasonable bounds to show that the $\mathbb{F}_p$ sketch of (Indyk, 2006) is
$(1/p\cdot \ln\rho_p)$-DP.  I like the flow of the exposition. This is a valuable contribution to the DP literature.

Weaknesses: The privacy parameter is not a constant but the DP guarantee should be a constant.
e.g., $\epsilon \in(0, 1])$. But this is not the case with this paper, rendering the paper
a little bit dangerous for the DP research community. There are other (perhaps minor) details about
proofs I'd like the authors to clarify. Also, the "real-world" dataset is not specified well (where is it from?). The description of the experiments on Page 9 should be a lot more detailed!

Their algorithm
provides, sometimes, vacuous privacy guarantees and inadequate accuracy.
We ideally want privacy of $\epsilon \leq 1$ but this seems to
only happen when $p$ is very close to 1.
If $\rho_p$ is the **multiplicative sensitivity** of the $p$-th frequency moment, then the
algorithm (the $\mathbb{F}_p$ sketch of Indyk) is $1/p\cdot \ln\rho_p$-DP, which is not a constant.

As described in several DP surveys, $\epsilon$ should be
a constant (ideally between 0 and 1).  But it appears this is not the case for the $\mathbb{F}_p$ sketch of
Indyk. I still think that certain improvements could help make this $\epsilon$ a constant.
As a baseline, the algorithm that uses sampling, to amplify privacy, can maintain polylogarithmic space complexity.
A $\epsilon$-DP algorithm (for $\epsilon\leq 1$)
that chooses, uniformly and at random, $\gamma n$ entries out of $n$ becomes $2\gamma\epsilon$-DP. Clearly,
if the algorithm uses polylog space for storing the samples, then this could serve as a baseline for
comparison for the $F_p$ sketch. (I recognize the authors also use privacy amplification by sampling but on top of the $F_p$ sketch.)

Also, the authors claim to give the first DP frequency estimation protocol with polylog space complexity
but this is clearly not the case. It is known, in the streaming literature, that for $p  > 2$,  the
space for $F_p$ estimation must be polynomial whereas polylogarithmic factors are achievable for
$p\leq 2$. And since there are DP algorithms for $p=0, 1, 2$, polylog space complexity has already
been achieved previously.


Other technical comments:

- In proof for Theorem 1 (page 4): in the first line after "to lower bound expression (1)", the
$(m-1)^{1-p}$ changes to $(m-1)^{p-1}$ in the next line. Is this a typo?

- The proof of Theorem 3 assumes that $\rho_p\geq 1$. Is this always the case?

- Proof of Theorem 5: How did you derive the variance of the sub-sampling process? Doesn't seem quite right to me.

## Post-Rebuttal Comments
The authors have addressed all my concerns.

**Summary Of The Paper:**

The authors show that the $\mathbb{F}_p$ sketch for frequency moments estimation is differentially private, as is,
for $p\in (0, 1]$. Other special cases for frequency moments estimation have been studied previously in the
literature (e.g., Choi et al., 2020, Smith et al., 2020, who study cases for $p = 0, 1, 2$).

For $p\in (0, 1]$, the authors build upon the $\mathbb{F}_p$ sketch of (Indyk, 2006) and show DP properties.
Specifically, they show, in Theorem 3,
that the $\mathbb{F}_p$ sketch of Indyk is $1/p\cdot \ln\rho_p$ where
$\rho_p$ is the **multiplicative sensitivity** of the $p$-th frequency moment.

The authors also present basic experimental evaluation of their algorithm (see Figure 3).
They also claim (wrongly, in my opinion) that their algorithm is the first with polylog space complexity
for $F_p$ estimation.

**Summary Of The Review:**

1) They provide reasonable bounds to show that the $\mathbb{F}_p$ sketch of (Indyk, 2006) is $(1/p\cdot \ln\rho_p)$-DP.  I like the flow of the exposition. This is a valuable contribution to the DP literature.

2) Inadequate privacy guarantee: In the literature, it is recommended that the privacy parameter be a constant. But instead it is $(1/p\cdot \ln\rho_p)$ which might provide no privacy at all for small $p$. [mostly addressed]

3) Experimental Evaluation: the experiments are rather sparse; also the "real-world" dataset is not specified. Where is it from? [addressed]

4) There might be some typos in the proofs. Also one calculation (Theorem 5) isn't fully specified. [now addressed]

---

> ### Author Response · Authors · 2021-11-13
> **Response to Reviewer Pbv7**
>
> We greatly appreciate reviewer Pbv7's positive and insightful feedback. The responses are provided inline as below.
>
> ***
>
> **Q:** The privacy parameter is not a constant and greater than 1.
>
> **A:** Thanks for the feedback. Given a lower bound of stream size, and the domain size and update step size, one can calculate the privacy parameter beforehand. We agree with the reviewer that it is not conveniently tunable but it can be made a constant. However, we argue that as the privacy guarantee comes for free, it is always better than no privacy.
>
> For $p\rightarrow 0$, the privacy guarantee is vacuous as the reviewer pointed out. However, as shown in Figure 3, when $p>0.3$, the privacy budget is smaller 4; when $p>0.6$, the privacy budget is smaller than 1. These are generally acceptable areas of $\epsilon$.
>
> Besides, it might be possible to amplify the internal privacy using tools like shuffling, sub-sampling or even some currently unknown amplification methods. We are actively investigating this direction now.
>
> ***
>
> **Q:** The “real-world” dataset is not specified well (where is it from)?
>
> **A:** Thanks for the comment! As specified in Section 4.1, the dataset is collected by TalkingData SDK. Each entry records the APP ID invoked in one access to TalkingData SDK. It can be viewed as a stream of incoming SDK accesses. Please let us know if there is any other information about the dataset that needs to be added.
>
> ***
>
> **Q:** The authors claim to give the first DP frequency estimation protocol with polylog space complexity but this is clearly not the case. It is known, in the streaming literature, that for p>2, the space for Fp estimation must be polynomial whereas polylogarithmic factors are achievable for p≤2. And since there are DP algorithms for  p=0,1,2, polylog space complexity has already been achieved previously.
>
>
> **A:** We apologize for the confusion due to our lack of elaboration. We state that “we make the first customized effort towards DP estimation of **fractional** frequency moments, i.e. **$p\in(0,1]$** with low space complexity.” Thus we only intend to claim our contribution to the case when $p\in(0, 1)$. We have also highlighted other works’ contribution to the cases of p=0, 1, 2 in the third paragraph of the Introduction section and the Related Work section.
>
> ***
>
> **Q:** In proof for Theorem 1 (page 4): in the first line after "to lower bound expression (1)", the (m−1)1−p changes to (m−1)p−1 in the next line. Is this a typo?
>
> **A:** We apologize for the confusion due to our lack of elaboration. The change from $(m-1)^{1-p}$ to $(m−1)^{p−1}$ is due to we multiply both the numerator and the denominator by $(m−1)^{p−1}$.
>
> ***
>
> **Q:** The proof of Theorem 3 assumes that $\rho_p \geq 1$. Is this always the case?
>
> **A:** The definition of $\rho_p$ is $\sup|\frac{\mathcal{M}(S)}{\mathcal{M}(S’)}|$ where S and S’ are exchangeable so $\rho_p$ must be greater than or equal to 1.
>
> ***
>
> **Q:** Proof of Theorem 5: How did you derive the variance of the sub-sampling process? Doesn't seem quite right to me.
>
> **A:** Thanks for the catch! The variance is wrong. We have fixed it and added the derivation of the variance to the proof of Theorem 5.
>
> ***
>
> Please let us know if you have any follow-up comments.

---

### Official Review · Reviewer_PGRe · 2021-11-02

**Correctness:** 4
**Technical Novelty And Significance:** 3
**Empirical Novelty And Significance:** 2
**Recommendation:** 8
**Confidence:** 3

**Main Review:**

The following are the strengths of the paper:

(+) The paper presents a solid theoretical result for a very (at least mathematically) natural problem.

(+) The paper shows that an existing algorithm for the frequency moment problem in the data stream literature is also \eps-DP. This is really nice since it shows that an existing algorithm is more powerful than what was already known.

(+) It seems like the notion of pure multiplicative sensitivity is new or perhaps more accurately not that well explored in the DP literature.

The main weakness of the paper are mostly from motivational/presentation for me:

(-) The paper starts off with some applications of p'th moment for p=1 and p=2 but from what I can tell there are no applications given for p\in (0,1), which is the range where the results in the paper are new. While it is mathematically natural to study the DP of this problem, one thing missing is to show (if possible) what _potential_ practical applications exists from having an \eps-DP algo for computing the p-th frequency moments for p\in (0,1).

(-) It would be useful to have a sense for why proving the results in the paper are hard? Or put another way what are the new technical ideas that are unique to this paper that was not present before that prevented previous works to solve the current problem? The paper states that the notion of pure multiplicative sensitivity is new but it also seems like the definition was already used for p=1 in Mir et al. (2011).

(-) From a theoretical point of view one potential drawback is that the result only holds for the turnstile model: i.e. where each update is positive. It seems like this is needed in proof of Thm 3 in the paper but it's not clear to me if this assumption is reasonable for potential applications of the results in the paper.

(-) The presentation of the paper is not ideal, though the issues I found (see details below) should be easily fixable.

(*) This is not really a concern but the paper did not have an ethics statement. While I can understand that the authors think that since the results in this paper are theoretical there are not ethical issues. While it is correct that there are no direct ethical issues, I would encourage the authors to think about potential mis-uses of the the algorithm by someone who potentially uses the results in this paper. Listing those in an ethical statement could be of use to others.

Below are some comments for the authors to improve on the presentation:

[Pg. 3, Def 2] \alpha (in the last line of def) is not defined.

[Algo 1] Is \delta needed? Also at this point "accuracy constraint" is not defined.

[Thm 1] When you say \M calculates F_p "accurately" what does it mean? Do you mean exactly? Another Q: is the \rho_M the same as \rho_p that is used in Thm 3? Finally the bound on \rho_M should have \le instead of =.

[Proof of Thm 1, para 1] The setting seems different from that on Def 3: why does this not affect the final bounds/results? Please justify.

[Pg. 6, line below first display eq] The claim after "Observe that" is not clear: the p-stability does not mention the constant of proportionality while the claimed equality uses the the fact that the constant of proportionality is \zeta^{-1/p}. Some justification of this would be nice.

[Pg. 6, proof of Thm 3] If there is space it would be nice to define the Laplace transform as well.

[Algo 2] \delta not required?

[Comparison with Kane et al. above Thm 5] When claiming that the space requirement of Algo 2 is close to Kane et al. is it for the same accuracy requirement on Kane et al.?


**Summary Of The Paper:**

This paper considers the problem of computing the p'th frequency moment for p\in (0,1] with \eps-Differential Privacy (or \eps-DP). I.e. it presents a randomized algorithm that given a stream of key-values pairs estimates the p-th frequency moment (of a vector indexed by the key and whose entry is the sums of values for the given key) that satisfies the property that given two key-value pair streams \S and \S' of the same size such that they differ in exactly on key-value pair, the probability that the output of the mechanism on \S and S' fall in the same measurable subset of the output space differ by a multiplicative e^{\eps} factor.

In fact the paper shows that an existing data stream algorithm due to Indyk based on p-stable distributions gives \eps-DP guarantees. Using standard techniques from DP literature, the paper shows how to amplify the privacy guarantee by subsampling. The space complexity of the algorithm in the paper is only a log factor away from the optimal space complexity of non-private frequency moment estimation.

The main technical ingredient is the use of what the paper calls _pure multiplicative sensitivity_ of the p'th frequency moment function (this notion was used for p=1 in Mir et al. (2011)). The bound on the pure multiplicative sensitivity follows from (fairly) simple algebra. Converting the sensitivity bound into \eps-DP guarantee reduces to showing a certain exponential depending on the p-th frequency moment is positive definite (which is done using the Laplace transform of the exponential).

The paper also presents some experimental results on the accuracy bounds for the algorithm on real and synthetic data.

**Summary Of The Review:**

Theoretically, the paper seems nice but it could be made stronger by providing motivations for the specific problem being studied in the paper, giving better intuition for what technically is new in this paper and by possibility solving the problem for the case when the updates can be negative. These reason being the paper a bit below a clear accept for me.

Post rebuttal comments
----------------------------

The authors have addressed most of my concerns about the paper. I'm still not super-convinced about the usefulness of the result for the entire range $p\in (0,1]$ since the entropy applications seems only to need values $p=1-\Delta$ for $\Delta\to 0$. However, _theoretically_ studying the whole range still makes sense. After the rebuttal the paper is more of a 7 for me.

---

> ### Author Response · Authors · 2021-11-13
> **Response to Reviewer PGRe (Part 2)**
>
>
> ***
>
> **Q:** From a theoretical point of view one potential drawback is that the result only holds for the turnstile model: i.e. where each update is positive. It seems like this is needed in proof of Thm 3 in the paper but it's not clear to me if this assumption is reasonable for potential applications of the results in the paper.
>
> **A:** Thanks for the comment! We assume there is a typo in the comment where “turnstile model” should be “cash register model”? Cash register model refers to a key-value stream whose items always have positive values. Strict turnstile model refers to a key-value stream in which the sum of values with the same key is always non-negative (i.e. values might be negative).
>
> We have extended the pure multiplicative sensitivity analysis to the strict turnstile model in Theorem 6 in Appendix A. Briefly, if we assume $M\ll n$, then the sensitivity is extremely close to the sensitivity in the cash register model (Figure 7). This analysis enables $F_p$ sketches to be used in a wider range of applications satisfying the strict turnstile model.
>
>
> ***
>
> **Q:** This is not really a concern but the paper did not have an ethics statement. While I can understand that the authors think that since the results in this paper are theoretical there are not ethical issues. While it is correct that there are no direct ethical issues, I would encourage the authors to think about potential mis-uses of the algorithm by someone who potentially uses the results in this paper. Listing those in an ethical statement could be of use to others.
>
> **A:** Thanks for the suggestion! We have added an ethics statement at the end of the paper.
>
> ***
>
> **Q:** Below are some comments for the authors to improve on the presentation:
>
> -- [Pg. 3, Def 2] \alpha (in the last line of def) is not defined.
>
> **A:** Thanks for the catch! It is a typo actually. $\alpha$ should be $p$.
>
> -- [Algo 1] Is \delta needed? Also at this point "accuracy constraint" is not defined.
>
> **A:** Thanks for the catch! We have removed those inputs from Algo 1.
>
> -- [Thm 1] When you say \M calculates F_p "accurately" what does it mean? Do you mean exactly?
>
> **A:** Yes, we have removed the word “accurately” to avoid possible confusion.
>
> -- Another Q: is the \rho_M the same as \rho_p that is used in Thm 3? Finally the bound on \rho_M should have \le instead of =.
>
> **A:** Yes, $\rho_M$ and $\rho_p$ are the same. We have replaced all $\rho_M$ in the paper with $\rho_p$ to avoid confusion. We have also changed $=$ to $\le$ as suggested. Thanks for the catch!
>
> -- [Pg. 6, line below first display eq] The claim after "Observe that" is not clear: the p-stability does not mention the constant of proportionality while the claimed equality uses the the fact that the constant of proportionality is \zeta^{-1/p}. Some justification of this would be nice.
>
> **A:** Thanks for the comment! We have added justification of the observation in the newest version.
>
> [Pg. 6, proof of Thm 3] If there is space it would be nice to define the Laplace transform as well.
>
> **A:** Thanks for the suggestion! We have added a citation to Laplace transform in the proof. We also polished the proof of Theorem 3 to make it easier to follow.
>
> [Algo 2] \delta not required?
>
> **A:** Thanks for the catch! We have removed all the unnecessary inputs to Algo 2.
>
> [Comparison with Kane et al. above Thm 5] When claiming that the space requirement of Algo 2 is close to Kane et al. is it for the same accuracy requirement on Kane et al.?
>
> **A:** Yes. Because the accuracy bound in Theorem 5 differs from vanilla $F_p$ sketch by an additive term, the asymptotic space complexity remains the same although the constant might increase. Thus, the space complexity is still asymptotically worse than Kane et al. by a logarithmic factor.
>
> ***
>
> Please let us know if you have any follow-up comments.

---

> > ### Comment · Reviewer_PGRe · 2021-11-19
> > **Question/Comments on Appendix result**
> >
> > Thanks for writing up the result for the strict turnstile model (and yes I did mean cash register model in my review: thanks for catching that!). There are some typos in the proof (e.g. it refers to proof of Thm 6 instead of Thm 1 and there is a reference to eq (7), which I did not see) but those are easy to fix.
> >
> > I have a comment on the comment on the bounds being very close when $M\ll n$. But this does not give the whole picture: the parameter $m$ also has to come into the picture. E.g. for fixed $n,M$ the bound in the appendix stays fixed while the bound in Thm 1 goes to zero.
> >
> > Also you state that turnstile model opens up new application: could you please be more specific in _which_ new applications are now in the picture with the result in the appendix?

---

> > > ### Author Response · Authors · 2021-11-20
> > > **Response to a follow-up question**
> > >
> > > Thanks for the insightful comment!
> > >
> > > 1. Thanks for the catch! We have fixed the reference to Thm 1. Equation (7) is the inequality above the reference.
> > >
> > > 2. We would like to clarify a confusion that when $m\rightarrow \infty$, the bound in Thm 1 does not converge to 0 because the exponent $\frac{p-1}{p}$ is negative. Thus, the term $m^\frac{p-1}{p}$ is typically small and that’s why we state that the bounds are close if $M\ll n$.
> > > 3. Sorry for the confusion due to our lack of elaboration. We are not claiming this will enable the algorithm to be used in new types of applications. Instead, we are saying that this will enable the sketch to be used in more application scenarios where the stream satisfies the strict turnstile model.

---

> > > > ### Comment · Reviewer_PGRe · 2021-11-20
> > > > **Mea culpa on large $m$**
> > > >
> > > > Oh, oops: sorry for the error on my part on my $m$ comment-- thanks for the correction. It makes sense now!

---

> ### Author Response · Authors · 2021-11-13
> **Response to Reviewer PGRe (Part 1)**
>
> We greatly appreciate review PGRe's positive and insightful feedback. The responses are provided inline as below.
>
> ***
>
> **Q:** The paper starts off with some applications of p'th moment for p=1 and p=2 but from what I can tell there are no applications given for p\in (0,1), which is the range where the results in the paper are new. While it is mathematically natural to study the DP of this problem, one thing missing is to show (if possible) what potential practical applications exists from having an \eps-DP algo for computing the p-th frequency moments for p\in (0,1).
>
> **A:** Thanks for the comment! A concrete instance is that the entropy of a data stream can be approximated with small space using $F_{1\pm\Delta}$ when $\Delta\rightarrow 0$ [1]. Our analysis provides an internal privacy guarantee for $F_{1-\Delta}$ sketch and to estimate the entropy we only need to add noise to $F_{1+\Delta}$ which can potentially greatly improve the accuracy under the same privacy budget. The same approach can be used to approximate G-test, a widely-used hypothesis testing whose downstream applications include causal inference, feature selection, cryptanalysis, etc.
>
> [1] Zhao, Haiquan, Ashwin Lall, Mitsunori Ogihara, Oliver Spatscheck, Jia Wang, and Jun Xu. "A data streaming algorithm for estimating entropies of od flows." In Proceedings of the 7th ACM SIGCOMM conference on Internet measurement, pp. 279-290. 2007.
>
> ***
>
> **Q:** It would be useful to have a sense for why proving the results in the paper are hard? Or put another way what are the new technical ideas that are unique to this paper that was not present before that prevented previous works to solve the current problem? The paper states that the notion of pure multiplicative sensitivity is new but it also seems like the definition was already used for p=1 in Mir et al. (2011).
>
> **A:** Thanks for the comment! First, the notion of pure multiplicative sensitivity is new (Mir et al. did not propose it; a similar but different analogue is from Dwork et al. 2015). We do agree that it is a natural extension of the common sensitivity notion. Second, the main challenging parts of the paper is the proof of Theorem 1 and Theorem 3. Specifically, given that stable distributions do not have close-form density functions, the analysis is extremely math-intense and involves real analysis which is beyond the typical math requirement for differential privacy analysis. Third, we argue that the idea, that $F_p$ sketches preserve DP as is, is an important contribution.

---

> > ### Comment · Reviewer_PGRe · 2021-11-19
> > **Question on applications of F_p for p\in (0,1]**
> >
> > Thanks for the clarification on potential applications to approximating entropy and G-test. Were there DP results known for these problems or would your results be the first DP results for these problems? If DP results for these problems already exists it would be good to see a quantitative comparison of your results with existing results.

---

> > > ### Author Response · Authors · 2021-11-20
> > > **Response to a follow-up question**
> > >
> > > Thanks for the follow-up questions! Our results are the first for these problems with polylogarithmic space. The only related work we are aware of is [1]. They propose a differentially private entropy estimation algorithm by adding Laplace noise to the best-polynomial estimator
> > > [2] which maintains a linear-size histogram.Thus, the results are not comparable because of the huge difference in space complexity.
> > >
> > > [1] Acharya, Jayadev, Gautam Kamath, Ziteng Sun, and Huanyu Zhang. "Inspectre: Privately estimating the unseen." In International Conference on Machine Learning, pp. 30-39. PMLR, 2018.
> > >
> > > [2] Wu, Yihong, and Pengkun Yang. "Minimax rates of entropy estimation on large alphabets via best polynomial approximation." IEEE Transactions on Information Theory 62, no. 6 (2016): 3702-3720.

---

### Official Review · Reviewer_P1Tg · 2021-11-03

**Correctness:** 3
**Technical Novelty And Significance:** 3
**Empirical Novelty And Significance:** Not applicable
**Recommendation:** 6
**Confidence:** 4

**Main Review:**

There has been a growing literature showing that different sketching techniques from the streaming literature preserve differential privacy. By now this is known for the Johnson-Lindenstrauss sketch of the $\ell_2$ norm, and the Flajolet-Martin distinct counts sketch. This paper adds to this line of work by showing that Indyk’s p-stable distribution sketches, which can be seen as an extension of the Johnson-Lindenstrauss sketch to $\ell_p$ norms for $p \in (0,2]$, also preserve differential privacy. The main idea is to first bound the *multiplicative* sensitivity of the $\ell_p$ norm, and then bound the ratio of the pdfs of the sketch for two neighboring streams in terms of this sensitivity.

The result is technically interesting, and provides further evidence that techniques from sketching are inherently privacy preserving. For this reason, I think the paper makes a nice conceptual contribution. Unfortunately, the authors don’t really give applications of their result. I would’ve liked to see an example scenario when would one want to publish a p-stable sketch of a stream? Since the randomness of the sketch itself must be kept hidden in order to protect privacy, this result is not applicable in a distributed setting, or in the pan-privacy setting, and I would like to see an example in which it is applicable.

Further detailed comments:

* Often these sketches are applied to the difference of two streams, in order to estimate the $\ell_p$ distance between them. In such cases the $v_i$ values might be negative. Can the analysis techniques here be extended to handle this situation?

* What is $\alpha$ in Definition 2?

* In Definition 3 and Theorem 1 you could just talk about functions rather than deterministic mechanisms.

* The RHS of (3) is also a direct application of Holder’s inequality.

* The authors claim that Theorem 2 was previously proven by Mir et al's PODS 2011 paper, but I cannot find any such result there. What theorem in that paper gives this result?

* In the proof of Theorem 2, please specify that the ratio of pdfs is a decreasing function *of $x$*.

* I can’t fully follow the proof of Theorem 3. Some comments:
- give a reference or prove that $\mathcal{D}_{p,1}(x)$ is increasing for $x \le 0$ and increasing for $x \ge 0$; also, I don’t understand how the upper bound on the ratio of pdfs follows from this fact;
- why is it sufficient to show (4), and how does this follow from the second derivative of $(1+u^{1/2})e^{-u^{1/2}}$?

* Instead of subsampling, can’t you improve the privacy loss parameter by padding the stream in order to increase $n$?

**Summary Of The Paper:**

The main result of the paper is that Indyk’s p-stable distribution sketch of the $\ell_p$ norm of a stream itself preserves differential privacy, without any additional randomization. The privacy loss parameter is a function of the length of the stream, the maximum value of any update, and the universe size. The privacy loss improves when the length of the stream goes to infinity with respect to the other parameters.

**Summary Of The Review:**

This paper makes an interesting technical contribution, giving further evidence that sketching techniques from the streaming literature preserve differential privacy without further randomization. The paper can be accepted, although it would be stronger if the authors gave applications of their results.

---

> ### Author Response · Authors · 2021-11-13
> **Response to Reviewer P1Tg (Part 2)**
>
> ***
>
> **Q:** why is it sufficient to show (4), and how does this follow from the second derivative of (1+u1/2)e−u1/2?
>
> **A:** We apologize for the lack of elaboration on this part of the proof. We have enriched the proof in the latest version to make it more coherent. The high-level idea is that we want to prove $F(u)=(1+u^{1/2})\exp{-u^{1/2}}$ is the Laplace transform of $f(t)=\frac{\exp{-1/(4t)}}{4\sqrt{\pi}}$: $F(u)=\int_0^\infty f(t)\exp(-ut) dt$. For the ease of algebra, we take the second derivative of both sides and scale $t$ and $u$ to get Equation (4). It is sufficient to prove (4) as LHS and RHS of the original equation (and their first derivatives) take the same value at $\infty-$.
>
> ***
>
> **Q:** Instead of subsampling, can’t you improve the privacy loss parameter by padding the stream in order to increase n?
>
> **A:** Thanks for the comment! By padding the stream with specially-keyed items, we can increase $n$ to push $\rho$ close to its lower bound $2^{2-2p}$ and thus improve privacy loss parameters. This should work when $n$ is much smaller than $M$ but won’t help too much if $n$ is already much larger than $M$.
>
> ***
>
> Please let us know if you have any follow-up comments.

---

> > ### Comment · Reviewer_P1Tg · 2021-11-18
> > **Still confused by the proof that F is the Laplace transform of f**
> >
> > What is $\infty-$? If $L$ and $R$ have the same second derivative, then they differ by at most a linear function. You seem to be claiming this linear function is $0$. Is your argument that this is the case because $\lim_{u \to infty}(L(u) - R(u)) = 0$?

---

> > > ### Author Response · Authors · 2021-11-18
> > > **Response to the follow-up question**
> > >
> > > Thanks for the follow-up question!
> > >
> > > (1) The definition of $\infty-$: $g(\infty-):=\lim_{x\uparrow\infty}g(x)$
> > >
> > > (2) Yes we are claiming the difference is 0. The reason is $L(\infty-)=R(\infty-)=0$ and $L'(\infty-)=R'(\infty-)=0$. The first equality eliminates the possibility of constant term difference and the second eliminates the possibility of linear term difference.
> > >
> > > We have updated the paper accordingly. Please let us know if there is any follow-up question.

---

> ### Author Response · Authors · 2021-11-13
> **Response to Reviewer P1Tg (Part 1)**
>
> We greatly appreciate reviewer P1Tg's positive and insightful feedback. The responses are provided inline as below.
>
> ***
>
> **Q:** I would’ve liked to see an example scenario when would one want to publish a p-stable sketch of a stream? Since the randomness of the sketch itself must be kept hidden in order to protect privacy, this result is not applicable in a distributed setting, or in the pan-privacy setting, and I would like to see an example in which it is applicable.
>
> **A:** Thanks for the comment!  We do agree that the confidentiality of the projection matrix is necessary to leverage the intrinsic privacy of $F_p$ sketch in a federated setting and it still has a long way to go. But in the centralized setting, our finding enables all existing applications of $F_p$ sketches to also provide a certain level of formal privacy guarantee for free. For example, the entropy of a data stream can be approximated with small space using $F_{1\pm\Delta}$ when $\Delta\rightarrow 0$ [1]. Our analysis provides a free privacy guarantee for $F_{1-\Delta}$ sketch and to estimate the entropy we only need to add noise to $F_{1+\Delta}$ which can improve the accuracy under the same privacy budget. The same approach can be used to approximate G-test, a widely-used hypothesis testing whose downstream applications include causal inference, feature selection, cryptanalysis, etc.
>
> [1] Zhao, Haiquan, Ashwin Lall, Mitsunori Ogihara, Oliver Spatscheck, Jia Wang, and Jun Xu. "A data streaming algorithm for estimating entropies of od flows." In Proceedings of the 7th ACM SIGCOMM conference on Internet measurement, pp. 279-290. 2007.
>
> ***
>
> **Q:** Often these sketches are applied to the difference of two streams, in order to estimate the ℓp distance between them. In such cases the vi values might be negative. Can the analysis techniques here be extended to handle this situation?
>
> **A:** Yes, the result in the paper can be directly used to handle this situation as long as the two streams both satisfy the cash register model or the strict turnstile model (Appendix A). The encoding $a$ (line 4 in Algo 1) can then be viewed as the difference between $a_1$ for the first stream and $a_2$ for the second stream. Using composition, we can get the privacy parameter for the final output.
>
> ***
>
> **Q:** What is α in Definition 2?
>
> **A:** Thanks for catching the typo! It should be $p$.
>
> ***
>
> **Q:** In Definition 3 and Theorem 1 you could just talk about functions rather than deterministic mechanisms.
>
> **A:** Thanks for the comment! We have changed Definition 3 as suggested.
>
> ***
>
> **Q:** The RHS of (3) is also a direct application of Holder’s inequality.
>
> **A:** Thanks for the comment! We have added that (3) is also an application of Holder inequality in the proof of Theorem 3 to help audiences from different backgrounds understand it.
>
> ***
>
> **Q:** The authors claim that Theorem 2 was previously proven by Mir et al's PODS 2011 paper, but I cannot find any such result there. What theorem in that paper gives this result?
>
> **A:** Thanks for the catch! Mir et al. also study how to make $\mathbb{F}_1$ sketch private but do not provide similar results. We have fixed the corresponding statements in the newest version.
>
> ***
>
> **Q:** In the proof of Theorem 2, please specify that the ratio of pdfs is a decreasing function of x
>
> **A:** Thanks for the comment! We have added specification why the ratio is decreasing to the proof of Theorem 2 in the newest version as suggested. Briefly, the derivative is always non-positive when $x\in(0, \infty)$.
>
> ***
>
> **Q:** give a reference or prove that Dp,1(x) is increasing for x≤0 and increasing for x≥0
>
> **A:** Wintner proved that symmetric stable distributions are unimodal (increasing in the negative half and decreasing in the positive half) when 0<p<2 in Section 11 of his paper “Cauchy's Stable Distributions and an "Explicit Formula" of Mellin” [2].
>
> [2] Wintner, Aurel. "Cauchy's stable distributions and an" explicit formula" of Mellin." American Journal of Mathematics 78, no. 4 (1956): 819-861.
>
> ***
>
> **Q:** also, I don’t understand how the upper bound on the ratio of pdfs follows from this fact;
>
> **A:** Because $\rho_p\geq 1$, $F_p^{-\frac{1}{p}}x \leq (\rho_pF_p)^{-\frac{1}{p}}x$ when $x\geq 0$ and takes equality when $x=0$. Thus, the maximum of the ratio is achieved when $x=0$.

---

### Official Review · Reviewer_49oY · 2021-11-06

**Correctness:** 4
**Technical Novelty And Significance:** 3
**Empirical Novelty And Significance:** 3
**Recommendation:** 8
**Confidence:** 3

**Main Review:**

The result is interesting and satisfying. The problem is important and it's good to have DP algorithms for it. The technical contribution is strong.

The main weakness is that the result only works for p in (0,1) and therefore feels incomplete.

**Summary Of The Paper:**

This paper proves that Indyk's classical Fp streaming algorithm for frequency moments satisfies the differential privacy (DP) requirements whenever p is in (0,1).

This leads to the first DP streaming algorithm for this problem with polylog space.

The new proof is based on a novel analysis of the sensitivity of this algorithm: how much does the output change when the input changes by a bit.



**Summary Of The Review:**

Strong and interesting results. Slightly incomplete, but still a good contribution.

---

> ### Author Response · Authors · 2021-11-13
> **Response to Reviewer 49oY**
>
> We greatly appreciate reviewer 49oY's positive and insightful feedback. The responses are provided inline as below.
>
> ***
>
> **Q:** The main weakness is that the result only works for p in (0,1) and therefore feels incomplete.
>
> **A:** Thanks for the feedback! As we discussed in the Conclusion section, $p\in(1, 2)$ is extremely hard to analyze as the monotonicity of the ratio of Fourier transforms in that region is extremely unstable and hard to analyze. We deem this as an extremely challenging and important future direction to pursue.
>
> ***
>
> Please let us know if you have any follow-up comments.

---

### Decision · Program_Chairs · 2022-01-20

**Decision:**

Accept (Poster)

**Comment:**

The reviewers agreed that this is a technically novel and interesting paper with results for a very natural problem and all voted for acceptance. The paper gives more evidence for the wide-ranging compatibility between the goals of sketching and of privacy.